# Dyeing with Hydrotalcite Hybrid Nanoclays and Disperse, Basic and Direct Dyes

**DOI:** 10.3390/ijms24010808

**Published:** 2023-01-03

**Authors:** Daniel López-Rodríguez, Jorge Jordán-Núñez, Jaime Gisbert-Paya, Pablo Díaz-García, Eva Bou-Belda

**Affiliations:** 1Departamento de Ingeniería Textil y Papelera, Universitat Politècnica de València, Plaza Ferrándiz y Carbonell s/n, CP 03801 Alcoy, Spain; 2Departamento de Ingeniería Gráfica, Universitat Politècnica de València, Plaza Ferrándiz y Carbonell s/n, CP 03801 Alcoy, Spain

**Keywords:** nanoclay dyeing, hydrotalcite, dye adsorption, dye desorption, direct dye recovery, removal of dyes, reactive dye recovery, TGA, FTIR, XRD

## Abstract

Textile effluents are among the most polluting industrial effluents in the world. Textile finishing processes, especially dyeing, discharge large quantities of waste that is difficult to treat, such as dyes. By recovering this material from the water, in addition to cleaning and the possibility of reusing the water, there is the opportunity to reuse this waste as a raw material for dyeing different textile substrates. One of the lines of reuse is the use of hybrid nanoclays obtained from the adsorption of dyes, which allow dye baths to be made for textile substrates. This study analyses how, through the use of the nanoadsorbent hydrotalcite, dyes classified by their charge as anionic, cationic and non-ionic can be adsorbed and recovered for successful reuse in new dye baths. The obtained hybrids were characterised by X-ray diffraction and infrared spectroscopy. In addition, the colour was analysed by spectrophotometer in the UV-VIS range. The dyes made on cotton, polyester and acrylic fabrics are subjected to different colour degradation tests to assess their viability as final products, using reflection spectroscopy to measure the colour attribute before and after the tests, showing results consistent with those of a conventional dye.

## 1. Introduction

The textile industry is one of the most water-intensive and polluting industries. On average, a textile processing unit of an average size generates about 125 L of effluent [1]. Analysing the standard effluent from the textile industry shows that there are significant amounts of chemical oxygen demand (COD), biological oxygen demand (BOD) and dissolved solids [2,3]. The pollution contained in these effluents is highly relevant [4] and it is, therefore, vital that these discharges are processed correctly [5]. Although there is a growing awareness of the problem and many governments are working on solutions for this, small industries cannot comply with the measures imposed and the survival of the textile industry itself is threatened. Thus, the concept of ecological textile dyeing processes that are viable for this industry has arisen.

Textile dyes can be classified in two different ways; one is based on their molecular structure and the second depends on the method of application to the textile materials [6]. The first way of classification is usually adopted by colour chemists, using expressions such as phthalocyanine, azo and anthraquinone. While the second method is commonly used by dyers and also by colourists in the dye manufacturing industries, using expressions such as soluble direct dyes for cellulosic materials such as cotton (CO), soluble basic dyes for acrylic fibres (PAN) and non-soluble disperse dyes for polyester fibres (PES). It should be noted that the categorisation of dyes according to their application is the main characteristic used for the colour index (CI).

Due to the increasing awareness of the impact of effluent pollution, researchers are investigating new cleaning and purification techniques. Among these new techniques is the use of nanoclays [7,8,9,10,11,12] as adsorbents for pollutants; also known as nanoadsorbents, they are capable of collecting dyes in textile effluents and separating them from the water. In previous works, several authors have demonstrated the adsorption capacity of these clays [13,14,15,16,17], although they have not reused these clays to make new dye baths. Some clays such as Laponite [18,19] can show desorption rates of around 20–40%. Another example where desorption has been achieved is with zeolites [20]. The literature also includes studies in which desorption is achieved with distilled water [19] or ethanol [21] and agitation, although none of these studies consider the possibility of a new dye.

In this work, Hydrotalcite (H) nanoclay is used as an adsorbent material. Hydrotalcite, Mg_6_Al_2_(CO_3_)(OH)_16_·4(H_2_O), is classified as a material of nanometric dimensions since its constituent lamellae have dimensions of less than 20 nm. Given the structure that it has, it falls into the “layered double hydroxides” (LDH) category. This layer has an SSA between 71 m^2^·g^−1^ and 104 m^2^·g^−1^ [22]. The aim of this research study is not only to trap the dye in the clay but also to be able to use the hybrid obtained in a new dyeing process. To this end, it is hoped to achieve a desorption in which the dye can be desorbed from the clay and used for the dyeing of a textile material that is susceptible to being dyed, taking into account the affinity of this type of dye and the textile fibre to be dyed in the process. Tests are carried out with dispersed dyes that have an affinity with PES, basic dyes that dye PAN and direct dyes that are capable of dyeing CO.

## 2. Results

### 2.1. Dye Adsorption Performance

After having carried out the adsorption process of the dyes with the clay, it can be seen that the values obtained were excellent. The adsorptions are above 95%, which corroborates with the expected behaviour of hydrotalcite [23,24,25,26,27]. Although in this work, dyes of all possible polarities, anionic, cationic and non-ionic, were used, and in all cases, almost complete adsorption was achieved, as had already been seen in other studies [28]. The difference in charge of the dyes was not reflected in these first results, as high adsorption was achieved for all three classes at these dye and clay concentrations (Table 1).

### 2.2. Hybrid Color Measurements

The values obtained in the calculation of the colour of the initial clay-dye hybrids (HDB199, HBY2 and HDR1) and those collected after the dyeing of each fabric (H2DB199, H2BY2 and H2DR1) are shown in Table 2 and are also shown in a colour diagram in Figure 1. For these calculations and representation, the reflectance values (λ) of each of the samples have been used. The instructions of the CIE 15:2004 standard [29] have been used to perform these calculations with a fully objective comparison of both absolute and relative colorimetric results. For all these calculations the illuminant used was the standardised illuminant D65 and it was necessary to use certain CIELAB colorimetric values that had been encoded by the CIE 1931 XYZ standard. Observing the CIE a*b* and CIE-Cab*L* diagrams, it can be seen that the clay has indeed achieved a very different colour tone from the original white, which is the first proof that the dye that is no longer in the aqueous solution has passed into the hydrotalcite.

The resulting colour of the hybrids correlates with the colour that the dyes had on their own, which means that the HDB199 sample is located in the area that corresponds to pure blues. The hybrid formed by the BY2 dye identified as sample HBY2 follows an angle and is located on the axis that is assigned to yellows. Looking at the third sample HDR1 in which the dispersed dye DR1 is found, it can be seen that it is assigned to red tones with a slight yellow influence.

Both the luminosity and the saturation of the HBY2 sample have higher values compared to the other two samples due to the characteristics of the yellow colour itself, which give it a characteristic level of luminosity and leave a certain limitation to the shade of colours that could be present if the concentration variables between the hydrotalcite and the dye itself were altered. However, in the HDB199 and HDR1 samples, a luminosity value of over 50 is observed, which implies that with this saturation the colour of the hybrids is chromatic and dark, making it possible to obtain a wider range of colours than in the case of the yellow sample by varying the clay/dye concentrations (Figure 1). Obviously, if mixtures of these three hybrids were made, given the purity of the colours in terms of their tonality and with these levels of saturation, a wide variety of colour ranges could be obtained.

### 2.3. X-ray Diffraction (XRD)

Regarding X-ray diffraction, the results and the comparison of hydrotalcite before (H) and after calcination (HC) can be seen (Figure 2). Looking at the H line one can see the diffraction peaks appearing at 11°, 23°, 34°, 39°, 46°, 60° and 61° which are, respectively, attributed to the crystal planes 003, 006, 012, 015, 018, 110 and 113 [30]. After calcination all these peaks disappear and diffraction peaks showing an amorphous Mg(Al)O_x_ mixed oxide structure can be seen in the HC line [31].

Previous studies using XRD analysis of the Mg-Al supports, show a typical hydrotalcite structure (2θ = 11.27; 34.46°, JCPDS n° 220700) and the XRD patterns in Figure 2 demonstrate the presence of mixed oxides due to the presence of Mg(Al^3+^)O of the MgO-periclase type in accordance with JCPDS n° 450946 [32]. By observing the traces of the calcined sample, we can see its correspondence with JCPDS n° 211152, which corresponds to the MgAl_2_O_4_ spinel structure [33]. Furthermore, XRD patterns of the dried Mg-Al (Figure 2) reveal two distinct crystalline phases: MgO (JCPDS 450946) and a hydrotalcite phase (JCPDS 220700).

By incorporating the HC into the dye solution, it is expected that the interaction produced between the clay and the dye will change the crystalline structure to some extent during the rehydration and reconstruction phase of the nanoclay. Figure 3 shows how this change occurs if the analysis is centred on the 003 plane at around 11°. Here, it can be seen how the peak of the uncalcined clay appears but then disappears after calcination. The effect is produced by the collapse caused by dehydroxylation in the basal space of the nanoclay layers and exfoliation of the basal space [34]. This process will be of great help for the dye to penetrate and become fixed between the layers. Simultaneously, the penetration of the dye and the hydration of the HC take place, which will result in the reconstruction of its structure due to its shape memory [35,36,37,38]. In the case of anionic dyes, they will be incorporated into the structure in place of other anions that were present before calcination such as -OH- and CO_3_^2−^.

The intensity of the band at 11° is explained by the fact that the dyes have an amorphous structure but the H has a crystalline structure [39,40,41]. As a consequence, the band will be more intense the more crystalline it is, i.e., the less dye it has, the less amorphous it will be. The HDB199, HBY2 and HDR1 curves show less intensity than H because they have a large amount of dye. However, the H2DB199, H2BY2 and H2DR1 samples have less dye and yet their intensity drops, which can be attributed to the fact that the hydrotalcite structure is being destroyed during the dyeing and desorption treatment, in a process similar to calcination, losing its crystalline form and becoming more amorphous.

### 2.4. Fourier Transform Infrared Spectroscopy FTIR-ATR Analysis

By making use of the Fourier transform, some very relevant information will be obtained for this study. On the one hand, by studying and comparing the graphs in Figure 4, it is possible to analyse the differences that exist between the uncalcined hydrotalcite and how the peaks change after calcination. There are two very characteristic bands of the nanoadsorbent at 1361 cm^−1^ which are assigned to the carbonate group -CO_3_^2−^ [42,43] and another one located in the range 3200–3600 cm^−1^ clearly centred at 3408 cm^−1^ and which is attributed to the stretching between the oxygen and hydrogen of the hydroxyl group -OH [27,41,43] of the water between the clay laminae. The vibrations produced by methylene CH_2_ [44] can be seen in the peaks at 2850 and 2918 cm^−1^. After the destruction of the clay structure due to calcination, these bands are practically flat. This leaves the clay with a positive polarity allowing the incorporation of new anionic groups to take the place of the electronegative groups (-CO_3_^2−^) that have left its structure. Additionally, we note the bands produced by the Al-OH bond at 767 cm^−1^, NO^3−^ produces another peak at 640 cm^−1^ and the bond between Mg and O shows a peak at 549 cm^−1^ [41,45].

Continuing with the analysis, Figure 5 shows the spectra offered by the three dyes and each of their hybrids after adsorption and after the dyeing process. Analysing these spectra, it can be seen that in all the hybrids the characteristic bands of the hydrotalcite are shown, as at 1361 cm^−1^ and there is the band corresponding to CO_3_^2−^ and in the range of 3200–3600 cm^−1^ with the peak centred at 3408 cm^−1^ which corresponds to the hydroxyl group -OH [46,47]. Once again, the appearance of these bands, which had disappeared during calcination, shows that the nanoclay has shape memory and reconstructs itself after rehydration. In the sample corresponding to the hybrid HDR1 and H2DR1, the band at 1361 cm^−1^ is more pronounced than in the samples HDB199 and HBY2, since due to the non-ionic nature of the dye, there has not been as much anionic substitution of CO_3_^2−^. In addition, this band remains constant after desorption in the case of the hybrid with disperse dye, but variations are observed in the graphs of the basic and direct dyes.

Continuing with Figure 5, it can be seen that the HDB199 sample and the DB199 dye have a band at 1100 due to the formates, acetates, among others [48]. Reviewing the literature, several studies indicate that the bands between 1400 and 1640 cm^−1^ are due to the benzenes in the dye [39,40,42,44]. On the other hand, the vibration produced by the sulphonate group [49] and due to the azo bond (RN_2_R’) [50] appear in the bands at 1030 and 1500 cm^−1^, respectively.

When analysing the BY2 graph, there is a zone between 1070 and 1434 cm^−1^ that can be assigned to the pyrone groups [46], which gives the basic character of the dye, and there is also a band at 1690 cm^−1^ that corresponds to carboxylic and phenolic groups [46,51,52]. The bands at 1227, 1370, 1478 and 2959 cm^−1^ are assigned to the aliphatic groups CH2 and CH3 and the bands at 752 and 765 cm^−1^ are due to the vibrations of the aromatic C-H groups [53].

Bands of the dye DR1 and its hybrids HDR1 and H2DR1 are shown at 1507 and 1341 cm^−1^ which are due to non-symmetric, and also symmetric stretching of nitrogen dioxide respectively [54,55]. Another peak at 1600 cm^−1^ is formed by aromatic -C=C- groups. Several other bands and peaks appear at 1386, 1142 and 858 cm^−1^, each of which is due to the bonding of N_2_, aliphatic amine -C-N- groups and by the C-H group close to nitrogen dioxide (NO_2_) [55].

For the three cases of the three hybrids and dyes, there is not much variation in the bands with respect to the hybrid before and after dyeing, except in the case of H2DB199 where a more intense CO_3_^2−^ band is seen. Comparing these bands with the initial bands of the dyes, it can be observed that the most relevant bands are those produced by the vibrations of the chemical structure of the hydrotalcite, although some characteristics of the dyes continue to appear at 1600 cm^−1^ in DR1, 1030 cm^−1^ in DB199 and some of the pyrone groups of BY2.

### 2.5. Colour Measurement of the Dyes

The main objective of this research is to carry out the dyeing of various textiles of different types by means of exhaustion dyeing using the hybrids obtained with nanoclays as the dyeing material. Once this objective has been satisfactorily achieved, the colouring obtained on the textile fabric is calculated and evaluated. Table 3 shows the results of the L*a*b* h and C*ab values and a representation in a chromaticity diagram can also be seen in Figure 6, thus being able to express the colour obtained in each dye in a quantitative way.

In view of the results obtained in the dye colour measurements, the results are totally in accordance with a conventional dye, as if the dye had been introduced directly into the bath and not a clay-dye hybrid. Take as an example the blue colour of TDB199 which takes on a slightly yellow tone due to the mixture with the cotton fabric which by its nature tends to have a yellowish tone, and this mixture produces this slightly greenish effect. On the other hand, the yellow TBY2 shows a very high luminosity above an L* of 75, something very characteristic of intense yellows. Moreover, the red TDR1 approaches very slightly orange tones also due to its union with the textile substrate, in this case, PES. It is worth noting the very high chroma level of yellow compared to the other two colours, which is situated at a value above 100, characteristic of yellow tones.

### 2.6. Colour Fastness

After dyeing the different textile materials used in this work, the next step was to check the capacity of that colour to remain fast in the fibre when subjected to different agents or external actions that may alter its union with the material. For this purpose, the so-called colour fastness tests are carried out. It should be remembered that hybrids have been used whose dyes have an affinity for the specific fibre that has been introduced into the dyeing process; the direct dye shows an affinity for cellulosic materials such as cotton [56,57,58], the dyes (cationic) have an affinity for the chemical structure of PAN (anionic) [59] and in the case of polyester, they are dyed by the disperse dyes as if they were an alloy [60,61,62] since when they reach the glass transition temperature (~68 °C), the dye and textile join in a similar way to that of metals when they are melted. Above the glass transition of polyester, both phases are mixed at the molecular scale. These described affinities allow the dyes to be deposited on the fibre in a first adsorption phase and then to penetrate completely into the fibre in the absorption phase. This allows the colour to be well fixed, although depending on the characteristics of the dye and the nature of the fibres, they will present better or worse results of colour fastness to different agents.

The results obtained for each sample and test are shown in Table 4. These results are expressed numerically according to the grey scale (GSc) from 1 to 5, where is 1 is low fastness and 5 is high fastness, all according to the standards of each test. All values have been calculated instrumentally with a reflection spectrophotometer according to UNE-EN ISO 105-A05 standard, for the colour degradation the calculations have been made using the formulas of Equation (1).
ΔE_F_ = [(ΔL*)^2^ + (ΔC_f_)^2^ + (ΔH_f_)^2^]^1/2^(1a)
GS = 5 − [ΔE_F_/1.7](1b)
GS = 5 − [log_10_ (ΔE_F_/0.85)/log_10_2](1c)

Equation (1). (a) Colour change for the determination of the greyscale index for degradation (b) GS if ΔE_f_ ≤ 3.4 (c) GS if ΔE_f_ ≥ 3.4.

Figure 7 shows graphically a summary of the results obtained for each dye. The results obtained are as expected given the characteristics of each dye and material. For example, direct dyes have a low fastness to wet treatments such as washing, reactive dyes have good fastness to the same wet treatments and disperse dyes have high fastnesses in general, due to the fact that the dye and the fibre form a kind of alloy that makes it very difficult for the dye to leave the fibre.

### 2.7. Scanning Electron Microscopy (SEM)

The topographical analysis of the dyed fabrics will reveal whether the colour of the fabrics is due to the surface deposition of the hybrid or whether there has been an absorption of the dye by the textiles. Thus, Figure 8 shows images of PES, PAN and CO, with which clearly show that there is hardly any hybrid residue on the surface of the fibres, but it is merely a residual quantity that in no case would be sufficient to give colour of the intensity and uniformity that the fabrics have. This analysis confirms and complements those carried out previously, showing without doubt that there has been a desorption of the dye, which has passed from the clay to the dye bath. 

### 2.8. BET Surface Area and Porosity Measurements

The BET surface areas, pore volumes and pore sizes are shown in Table 5. The results of hydrotalcite before and after calcination as well as after adsorption of the different dyes have been analysed and compared. The results show that surface area, pore volume and pore size increase after calcination. This is due to the fact that during calcination the clay structure opens up. Previous studies claim that these changes are due to outgassing for the catalysts produced by the decomposition of the hydrotalcite gases into their hydrated phases [63,64,65]. On the other hand, the samples that have adsorbed the dye have very similar values to those of the clay before calcination.

## 3. Discussion

In this work, a new method of dyeing textile fabrics by reusing the dye-hydrotalcite hybrids formed by the adsorption of dyes obtained from discharged textile wastewater was successfully carried out. The loading process of the nanoclay was reversed to achieve the desorption of the dyes into the dye bath. This novel contribution has been analysed using colour measurements, and SEM analysis and by subjecting the specimens to different colour fastness tests. After desorption, there is still some dye left in the clay, which would allow the dyeing process to be repeated to obtain less intense colourings or to vary the clay concentrations to match the tone of the dyes used. In any case, these variables will be the subject of further research to continue this line of investigation.

The dyed fabrics have been subjected to various fastness tests, which have given fairly standard results, taking into account the type of dye and the dyed textile material. For example, it is usual to find low fastness values in wet treatments for direct dyes and high degrees of fastness in disperse dyes, since the dyeing of the latter is like an alloy produced between the dye and the polyester.

The already-known adsorption capacity of hydrotalcite for anionic pollutants has been confirmed and its ability to adsorb non-ionic and cationic elements has also been confirmed. XRD and FTIR analysis reaffirms the reconstruction capacity of this nanoclay thanks to its shape memory and the presence of the dyes incorporated in its structure. The X-ray analysis shows the increase in the amorphous zone due to the degradation of the clay with the desorption process, and the infrared analysis shows significant peaks of amino or sulphonate groups that confirm the hypotheses of dye adsorption and reconstruction of the nanoclay in the process.

## 4. Materials and Methods

### 4.1. Materials

In this work, three dyes with different characteristics were used to test the method when there is a different polarity of the dye and the dyeing conditions are different. An anionic dye of the direct type called direct blue 199 CI 74180 (DB199), another cationic dye classified as basic with the denomination Basic Yellow 2 (BY2) C.I. 41000 and finally a non-ionic dye without polarity of the disperse class referred to as Disperse Red 1 CI 11110 (DR1) were used. Their structures are shown in Figure 9.

Hydrotalcite Mg_6_Al_2_(CO_3_)(OH)_16_·4(H_2_O) [66,67,68] of Sigma Aldrich Gillingham (U.K), was the adsorbent used. There are different methods that can be used to achieve the adsorption of dyes in this kind of nanoclay. For this study, calcination was selected following the method described by Dos Santos R.M.M. [27]. This method, called calcination, consists of introducing the clay in a kiln at 600 °C for 3 h in order to destroy the H structure and reduce the presence of certain anions such as CO_3_^2−^ carbonate, which will allow for the incorporation of new negatively charged compounds in a later reconstruction phase during hydration thanks to the shape memory of this mineral. After this calcination process, it is considered to be calcined hydrotalcite (HC). Its lamellar structure will allow it to adsorb and fix other elements that are not anionic, so this material is not exclusive for adsorbing negatively charged pollutants.

Figure 2 shows the change in the structure of the H in the three cases described, prior to calcination, after being subjected to 600 °C for 3 h and after hydration and reconstruction. It is worth noting that in the last SEM and TEM images (Figure 10), it can be seen how the clay structure has been reconstructed [69]. Within Figure 10, image “b” shows how the structure has been destroyed by calcination and in figure “c” it can be seen how the structure has been recovered, being very similar to the original “a”. In addition, in image “e” it can be seen how the layers are further apart after calcination, which improves the adsorbent capacity as it is more likely that the adsorbate can be introduced between these layers.

### 4.2. Synthesis Methods

After a dyeing process by depletion, the dye remains in the dye bath; just as after the adsorption effect, the dye that has not been adsorbed may remain [47,70]. In order to determine the amount of dye in the form of g·L^−1^ concentration in each of these cases, simple regression models by Lambert-Beer [71] are used beforehand. Starting from various dilutions of the dyes at controlled concentrations, the adsorbance can be measured using a transmission spectrophotometer and the equations given in Table 6 can be obtained.

The first objective to be met in the study is to achieve the maximum possible dye adsorption in order to leave the water completely clean. To assess the adsorption capacity of the hydrotalcite, 4 L of a solution of each of the dyes to be studied was prepared at a concentration of 1 g·L^−1^. The next step was to introduce the nanoclay into these solutions. The amount introduced was 3 g·L^−1^. Once the mixture is prepared, it is subjected to agitation using a magnetic stirring system in which the maximum possible speed is applied at 1600 r.p.m. for 2 h. The speed is then changed to 500 r.p.m. for a further 22 h [72]. In the first 2 h, the aim is to achieve penetration of the dye with maximum centrifugal force, but then the speed is lowered to ensure that the dye does not come out of the clay again and remains as stable as possible, allowing its accommodation in the structure of the reconstructed nanoclay.

The hybrid formed by the hydrotalcite and the dye is then separated from the water. To do this, the solution is filtered using filter paper and all the aqueous part is separated from the solid part by gravity so that the hybrid can be collected in solid form after 48 h. Samples are then taken from the water that has fallen by gravity. This water is taken to the transmission spectrophotometer where, with the absorbance reading and using the equations in Table 5, the concentration of dye that still remains in the solution and has not been adsorbed is calculated [73,74]. On the other hand, the solid hybrid is freeze-dried [26,27] in order to extract all the water and avoid agglutinations that could occur during drying in the oven. In this way, the hybrids identified as HDB199, HBY2 and HDR1 are obtained for the union of the dyes DB199, BY2 and DR1, respectively.

The desorption process is described as a phenomenon in which there is a transfer of the dye from the adsorbate in the solid to the liquid phase [75,76]. Several models [77,78,79] explain different theories involving isotherms on how this desorption occurs, describing them as non-ideal and reversible adsorption/desorption systems. Another describes a system in which there is not always an interaction between neighbouring active sites due to the non-homogeneity of the nanoadsorbate, and therefore, no homogeneous adsorption. All these theories give an insight into the true nature of the adsorption/desorption process. The authors Momina, Shahadat Mohammad, and SuzylawatiIsamil [80] explain a desorption model for methylene blue (MB) by first subjecting the hybrid to high temperatures to weaken the bonds and then using various solvents such as HCl, ethanol, and nitric acid or acetone. They argue that any one of these phases alone is not sufficient to produce good desorption results.

In this study, a simultaneous desorption-dyeing process is proposed, in which, based on the theory that temperature weakens the bond between the clay and the dye [80], subsequently taking advantage of the dye-fibre affinity and the dyeing process commonly used so that the dye migrates completely from the hydrotalcite to the textile fibre (Figure 11). Furthermore, in the model of this work, heat is applied by convection and not by radiation as in the model of the authors Momina, Shahadat Mohammad, and SuzylawatiIsamil, as this heat is more effective at reaching more areas of the clay and is also more energetic.

The clay-dye hybrid is then used as a dyeing material for dyeing by exhaustion, using a bath ratio of 1/40. For the dyeing of a 100% cotton (CO) openwork fabric with a grammage of 135 g·m^−2^, 25 yarns·cm^−1^, 22 weft·cm^−1^ of plane weave openwork fabric, the direct dye hybrid HDB199 is used, 40% s.p.f. of clay + dye, 20 g·L^−1^ of sodium sulphate and three drops of a wetting agent are added to the dyeing bath to submit it to the dyeing process for 60 min at 100 °C, obtaining the dyed fabric referenced as TDB199. For the dyeing of a 100% polyester fabric (PES) 200 g·m^−2^, 13 yarns·cm^−1^, 52 weft·cm^−1^ of plain weave was introduced in a bath containing 40% s.p.f. of clay + HDR1 dye, for 60 min at 140 °C in a closed machine with 1 g·L^−1^ ammonium sulphate, 0.5 g·L^−1^ Dekol SN dispersant after previously adjusting the pH to 4.5–5 with acetic acid, thus obtaining the sample with reference TDR1. The last dyeing was on a fabric of 100% acrylic composition (PAN) a weft knitted fabric with eight rows per centimetre and nine columns per centimetre forming an English knitted weave in whose bath 40% s.p.f. of clay + HBY2 dye, acetic acid 2% s.p.f., 20 g·L^−1^ of sodium sulphate was added and processed for 40 min at 100 °C, thus obtaining the dyed fabric referenced as TBY2. The dyeing of the polyester fabric was carried out in a closed machine due to the temperatures above 100 °C that must be used. The apparatus used was the Testherm type 9S from the manufacturer Talcatex S.A, San Sebastian de los Reyes (Spain). Conversely, the dyeing of acrylic and cotton fibre was carried out in the open machine referenced as Open Bath dye Master from the manufacturer Paramount S.A, Geneva (Switzerland). All the fabrics were washed after dyeing to eliminate any remaining dye that was not fixed to the fibres.

After the dyeing process described above, the clay that was in the dye baths was collected again to assess the desorption that it has undergone. For this purpose, the dye baths are separated from the dyed fabrics and are again filtered by gravity with filter paper, as was conducted in the previous process. The clay is analysed again after dyeing to assess its colour change and other characteristics that may have been altered after this process. From each of the hybrid samples HDB199, HBY2 and HDR1, new clay-dye hybrids are obtained from the remainder collected after the dyeing, respectively, referred to as H2DB199, H2BY2 and H2DR1. 

### 4.3. Characterisation

The colour measurement of the obtained hybrids was studied using the Jasco V-670 double UV-VIS/NIR spectrophotometer. Measurements were carried out in the range of 2700–190 nm at a frequency of 0.5 nm. The Jasco V-670 is equipped with a double-grating monochromator. The first grating monochromator is used for the UV-VIS region serving 1200 grids-mm^−1^ which is equipped with detectors based on a photomultiplier tube. On the other hand, the second grating is used for the rest of the spectrum studied, i.e., for the IR infrared region, but this time with 300 grids-mm^−1^ and using a PbS detector. Both gratings are equipped with an automatic system that allows them to adapt to changes in wavelength. The light sources were a halogen lamp (330–2700 nm) and a deuterium lamp (190–350 nm). The CIE-1964 observer was used under the D65 illuminant, reflectance factors were also applied to obtain optical values for comparison [81].

A scanning electron microscope (SEM) model PHENOM (FEI Company, Eindhoven, The Netherlands) was used to perform the topographical analysis of the surface of the samples. It was operated at an acceleration of 5 kV. The previous sample preparation consists of sputtering with a palladium/gold alloy with an EMITECH sputter coater mod. SC7620 (Quorum Technologies Ltd., East Sussex, UK). As the coating thickness is only 5–7 nm it will not alter the readings. For TEM imaging, a JEOL model JEM-2010 transmission electron microscope was used. The image acquisition camera is a GATAN model ORIUS SC600. It is mounted on an axis with the microscope at the bottom and is integrated into the image acquisition and processing software GATAN DigitalMicrograph 1.80.70 for GMS 1.8.0.

The clay-dye hybrids were subjected to infrared spectrophotometer analysis in order to calculate the Fourier transform (FTIR). Due to the characteristics of the material to be analysed, the horizontal attenuated total reflection technique (FTIR-ATR) was used using a ZnSe prism. The instrumentation used for the readings was the Jasco FTIR 4700 IRT 5200 spectrophotometer with a DTGS detector sensor. It was necessary to use a pressure accessory to obtain a uniform reading on each of the samples. The spectrophotometer worked at a resolution of 4 cm^−1^ and scanned 64 scans.

Continuing with the characterisation of the hybrids obtained from the clay-dye, these samples were subjected to X-ray diffraction (XRD) tests [82,83] in order to analyse their behaviour, especially the changes in their lamellar structure during the calcination process and reconstruction during rehydration. Special attention is paid to the basal space between the hydrotalcite lamellae, which produces the adsorption of both ions and other non-ionic substances. For this purpose, the RD bruker D8-Advance (Bruker, Billerica, MA, USA) with a Göebel mirror (power: 3000 W, voltage: 20–60 kV and current: 5–80 mA) was used. The analysis was performed in an oxidising atmosphere at an angular velocity of 1°/min, STEP 0.05°, and an angular sweep of 2.7–70°. The diffraction patterns were indexed by making a comparison with the JCPDS files.

The dyeing samples of the three dyes were subjected to different fastness tests to assess their correct dyeing and subsequent behaviour. In order to check their fastness to washing, each of the samples was subjected to washing according to the UNE-EN ISO 105-C06:1994 standard using the Linitest described in this standard. The test carried out was the A1S test described in the standard at a temperature of 40 °C for 30 min and with a bath volume of 150 mL. The pH was not adjusted and 10 steel balls were added to generate an abrasive action. Tests for colour fastness to ironing were carried out according to UNE-EN ISO 105-X11 using a pressure plate. The tests were carried out in wet, damp and dry conditions as stated in the standard. The ironing time for all samples was 15 s at a temperature of 200 °C for PAN and PES fabrics, although for CO fabrics it was conducted at 150 °C, as cotton may yellow at higher temperatures. To assess the colour fastness to rubbing, the Crockmeter was used according to the UNE-EN ISO 105-X12 standard. This test was carried out wet and dry as described in the standard.

Colour degradation and discharge were measured instrumentally using a Minolta CM-3600d reflection spectrophotometer in the range 360–740 nm with a step of 10 nm according to UNE-EN ISO 105-A05 for degradation and UNE-EN ISO 105-A04 for discharge. The results are expressed according to the grey scale as stated in the aforementioned standards.

BET analysis was performed to measure the surface area, pore volume and pore size using nitrogen adsorption and desorption values at −196 °C on a Micromeritics ASAP-2020. The samples are first degassed in a vacuum atmosphere at temperatures between 150 °C and 200 °C so as not to carbonise any elements in the sample [64,65].

## Figures and Tables

**Figure 1 ijms-24-00808-f001:**
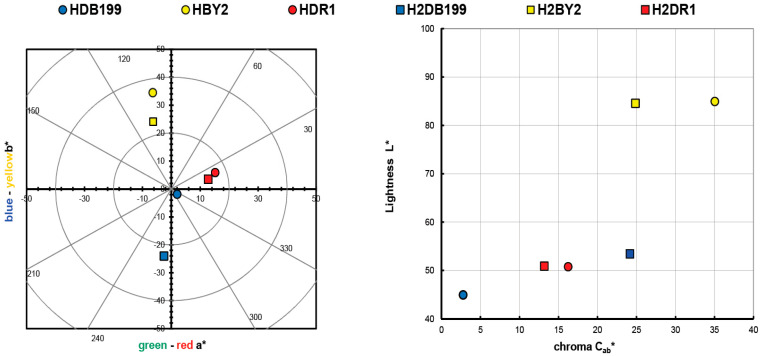
Graphic CIELAB plots for hybrid pigments synthesised using the D65 illuminant and the CIE-1931 XYZ standard observer. (**Left**): CIE-a*b* color diagram; (**right**): CIE-Cab*L* color chart.

**Figure 2 ijms-24-00808-f002:**
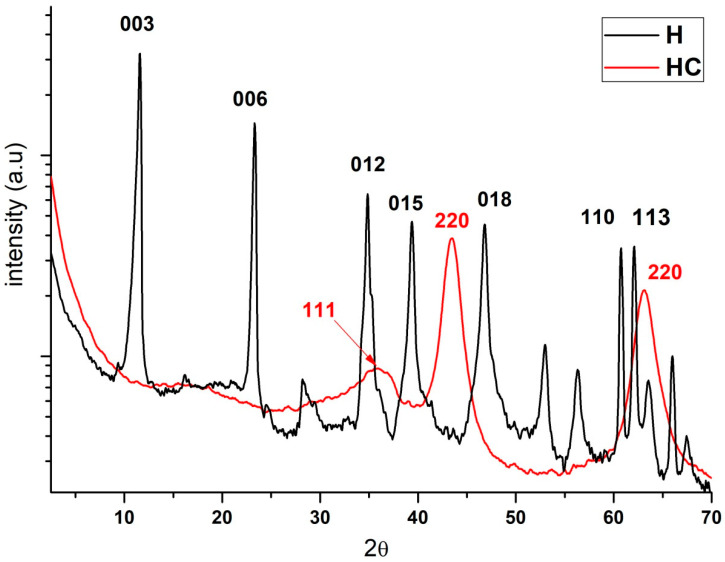
Diffraction patterns of the hydrotalcite without calcining (H), the hydrotalcite after the calcination at 600 °C for 4 h.

**Figure 3 ijms-24-00808-f003:**
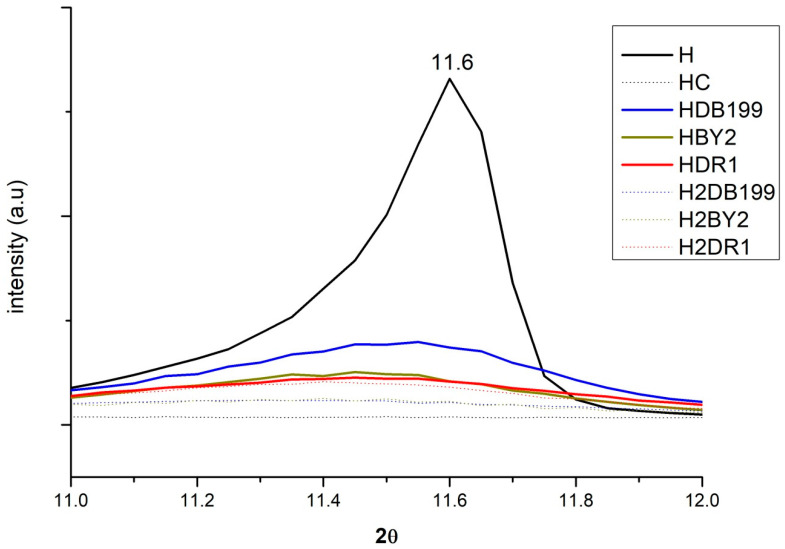
XRD for Hydrotalcite, Hydrotalcite calcinated at 600 °C for 4 h., samples HDB199, HBY2, HDR1, H2DB199, H2BY2 and H2DR1 in the range of 10° to 12.5°.

**Figure 4 ijms-24-00808-f004:**
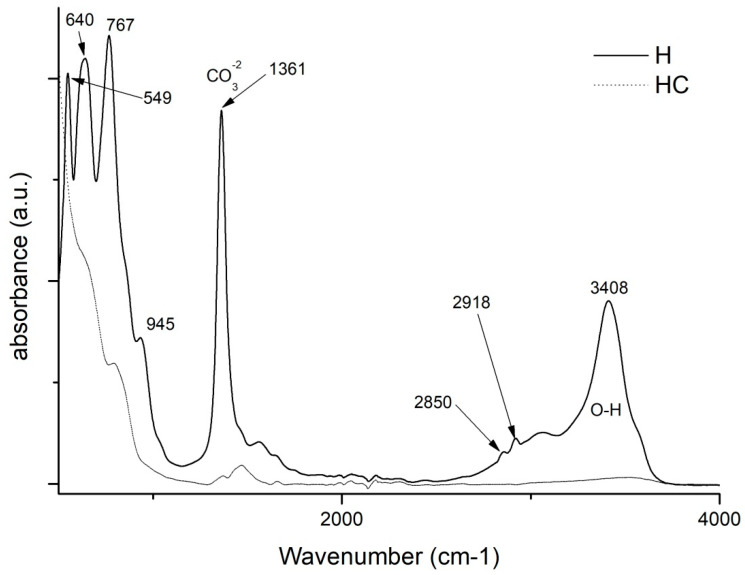
FTIR comparison of Uncalcined Hydrotalcite (H) and Calcined Hydrotalcite (HC).

**Figure 5 ijms-24-00808-f005:**
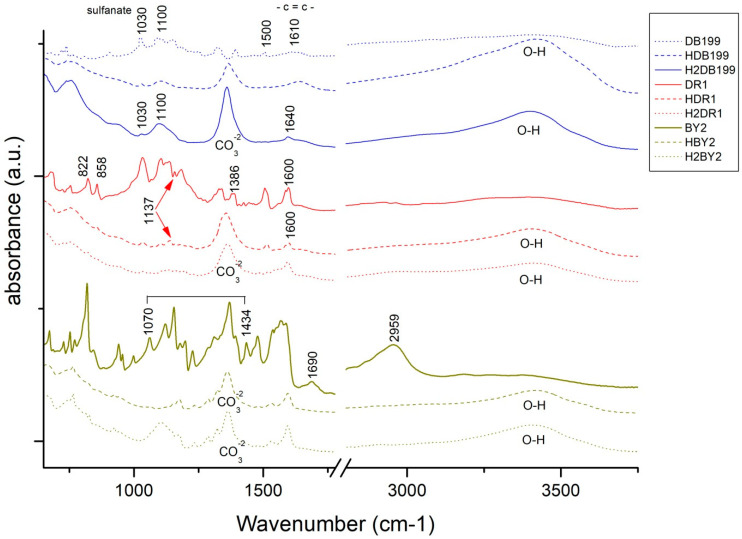
FTIR of DB199, BY2, DR1 and samples HDB199, HBY2, HDR1, H2DB199, H2BY2 and H2DR1.

**Figure 6 ijms-24-00808-f006:**
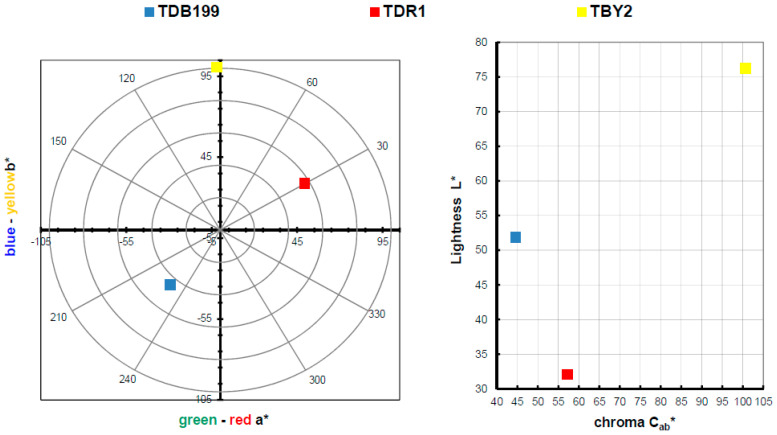
Graphic CIELAB plots for hybrid printed using the D65 illuminant and the CIE-1931 XYZ standard observer. (**Left**): CIE-a*b* color diagram; (**right**): CIE-Cab*L* color chart.

**Figure 7 ijms-24-00808-f007:**
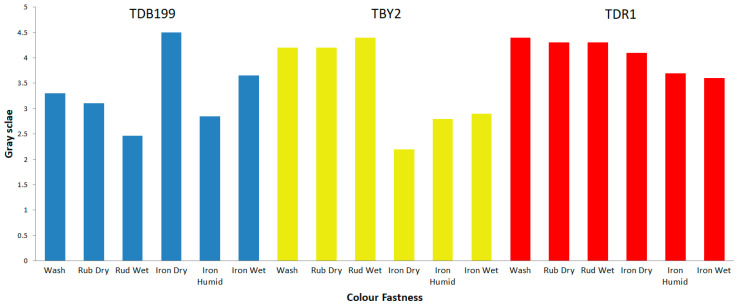
Colour fastness values by greyscale index.

**Figure 8 ijms-24-00808-f008:**
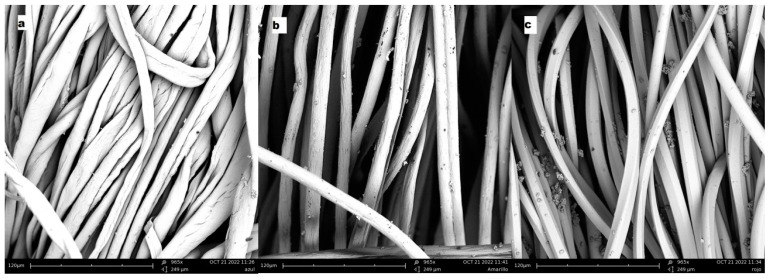
(**a**) SEM sample TDB199, (**b**) SEM sample TBY2, (**c**) SEM sample TDR1.

**Figure 9 ijms-24-00808-f009:**
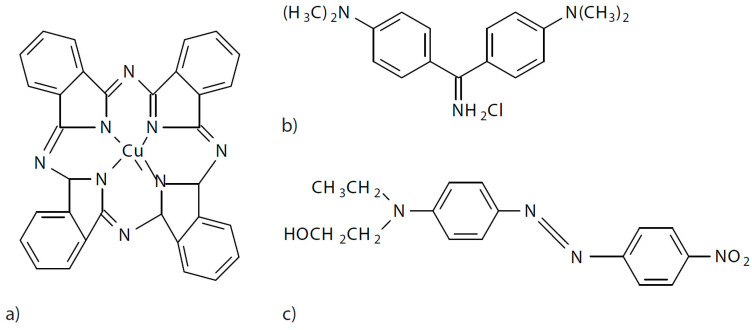
(**a**) Direct Blue 199, (**b**) Basic Yellow 2, (**c**) Disperse Red 1.

**Figure 10 ijms-24-00808-f010:**
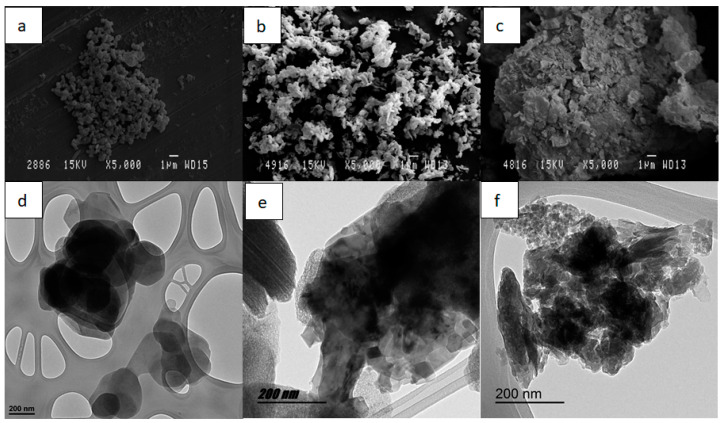
SEM micrographs of different H samples: (**a**) H original (**b**) HC calcinated (**c**) HC reconstructed. TEM micrographs of different H samples: (**d**) H original, (**e**) HC calcinated (**f**) HC reconstructed.

**Figure 11 ijms-24-00808-f011:**
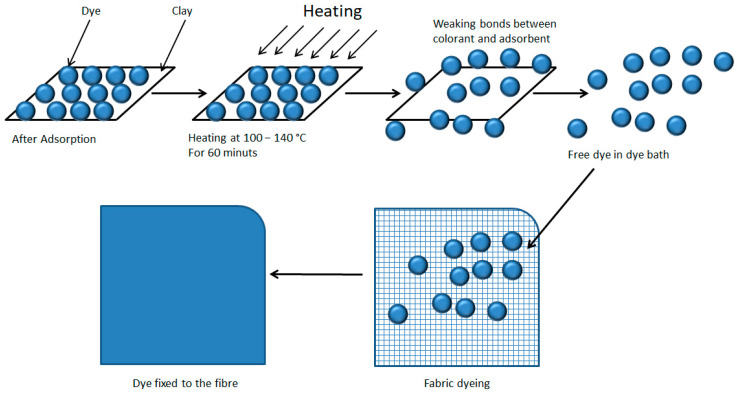
Dye desorption and dyeing of the textile fibre.

**Table 1 ijms-24-00808-t001:** Difference in concentration after HC adsorption.

Sample	Polarity	Initial Conc. g·L^−1^	Final Conc. g·L^−1^	Ads (%)
HDB199	Anionic	1	6.57 × 10^−4^ ± 3.2 × 10^−8^	99.316
HBY2	Cationic	1	6.22 × 10^−4^ ± 1.1 × 10^−8^	99.949
HDR1	Non-ionic	1	6.32 × 10^−3^ ± 2.9 × 10^−9^	98.122

**Table 2 ijms-24-00808-t002:** Values L*, a*, b*, C*_ab_ and h of each hybrid.

Sample	L*	a*	b*	C*_ab_	h
HDB199	44.94	2.03	1.88	2.77	317.16
HBY2	84.94	6.41	34.46	35.05	100.53
HDR1	50.77	15.14	5.86	16.24	21.17
H2DB199	53.43	−2.53	24.03	24.17	263.98
H2BY2	84.59	6.26	24.07	24.87	104.57
H2DR1	50.89	12.73	3.45	13.19	15.16

**Table 3 ijms-24-00808-t003:** Values L*, a*, b*, C*_ab_ y h of each dyeing.

Sample	Dye Shade	L*	a*	b*	C*_ab_	h
TDB199	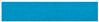	51.86	−29.17	−33.79	44.63	229.20
TBY2	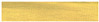	76.23	−1.97	100.61	100.63	91.12
TDR1	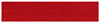	32.05	49.45	28.75	57.20	30.18

**Table 4 ijms-24-00808-t004:** Colour fastness values by greyscale index.

	Colour Fastness					
	Wash	Rub		Iron		
Sample		Dry	Wet	Dry	Humid	Wet
TDB199	3–4	4–5	4	4–5	4	3–4
TBY2	4	4	4–5	2	3	3
TDR1	4–5	4–5	4–5	4	4	3–4

**Table 5 ijms-24-00808-t005:** BET surface areas, pore volumes and average pore sizes.

Sample	Surface Area (m^2^/g)	Pore Volume (cm^3^/g)	Average Pore Size (nm)
H	114.3	0.21	10.07
HC	239.6	0.37	18.7
HDR1	102.2	0.15	11.8
HDB199	93.1	0.22	10.15
HBY2	98.5	0.18	11.25

**Table 6 ijms-24-00808-t006:** Lambert-Beer line equations and R^2^.

Dye	Equation	R^2^
Direct Blue 199 (DB199)	y = 21.784 x − 0.015	0.9982
Basic Yellow 2 (BY2)	y = 18.023 x − 0.0112	0.9972
Disperse Red 1 (DR1)	y = 25.411 x − 0.0244	0.9989

## Data Availability

Not applicable.

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
