# Peer review of "Dyeing with Hydrotalcite Hybrid Nanoclays and Disperse, Basic and Direct Dyes"

_ijms, 2023, doi:10.3390/ijms24010808_

Round 1
Reviewer 1 Report
Dyeing with hydrotalcite hybrid nanoclays and disperse, basic and direct dyes
General comment : please do not use acronyms in the captions of the tables and the figures.
· Introduction :
o What is an average size ? Could you be more specific: in the world, in Europe,… Quantity produced?
o 125 L/second or / hour ?
o If possible, I would have been interested in knowing the standard concentration
o Nanoclays : could you be more specific ? As far as I know, standard clay (Kaolinite) is made of platelets. The platelets are 20-80 microns lateral size. By dispersion/exfoliation it might be possible to get thin layer of platelets of several nanometers. Based on this knowledge, what do you mean by nanoclay ? What is the specific surface area of a clays versus a nanoclays ?
· Use the hybrid obtained in a new dyeing process. You mean using the clay as a dye carrier i.e. adsorption of the dye on the clay and desorption of the dye from the clay to use again the dye ? Þ I read the figure 11 and understand this at the end of the paper. Use the hybrid obtained in a new dyeing process, i.e. desorbing the dye from the clay in order to reuse it.
· Base on the table 1, the adsorptions are above 98% (not 95%). I have some doubts on your ability to measure a concentration of 6.57 10-4 gL-1, perhaps 6.610-4 gL-1 Ads% 99.316 : 99.3% ? What are the standard deviation of your measures? (adding “±” would give value to your results)
· Table 2 : same remark than table 1 Þ accuracy of the data would be interested…
· Line 122 : there is an extra « . »
· Figure 2. Calcination at which temperature and for how much time? (add: 600°C for 3 hours)
· 134 : It is assume that the dye becomes fixed between the layers. This is an assumption, it is difficult to check whether the dye in on the surface of the platelets or between the platelets ? It seems that the figure 11 shows that you believe more in a surface bonding than an intercalation. If it is the case, you just write adsorption on the clay in the text…
· Figure 3 : calcination temperature and temperature dwell. (add: 600°C for 3 hours)
· 165 : NO3-
· 176 : Nanoclay has a shape memory ? You mean that the bonds between the platelets are strong enough to reaggregate the platelets after exfoliation ?
· 132 : CO32
· 183 : formats, acetate (formats ?)
· 184 : Azo bond (RN2R’)
· 245-47 : Glass transition could you give an idea of the temperature of it ? You made a comparison with alloy ? Both the polyester and the dye have a specific glass transition and above the glass transition of poylester (~68°C ?) both phase are mixed at the molecular scale.
· 272 : surface deposition of the hybrid add: (i.e. adsorption of the hybrid on the fiber) Did I understand well?

Author Response
- What is an average size? Could you be more specific: in the world, in Europe,… Quantity produced?
- 125 L/second or / hour ?
- If possible, I would have been interested in knowing the standard concentration
- We refer to the fact that there are several production dimensions used in the textile industry, there are dyeing machines that may use 50 L, others 150 L, others 250 L, etc. Since not all of them can be explained by taking a single measurement, the closest thing to the average are the values given for 125 L for a dyeing process.
- Nanoclays: could you be more specific ? As far as I know, standard clay (Kaolinite) is made of platelets. The platelets are 20-80 microns lateral size. By dispersion/exfoliation it might be possible to get thin layer of platelets of several nanometers. Based on this knowledge, what do you mean by nanoclay ? What is the specific surface area of a clays versus a nanoclays ?
- In the introduction it has been added: Hydrotalcite, Mg6Al2(CO3)(OH)164(H2O), is classified as a material of nanometric dimensions since its constituent lamellae have dimensions of less than 20 nm. Given the structure that has, it falls into the “layered double hydroxides” (LDH) category. This layer has an SSA between 71 m2·g−1 and 104 m2·g−1
- Use the hybrid obtained in a new dyeing process. Youmean using the clay as a dye carrier i.e. adsorption of the dye on the clay and desorption of the dye from the clay to use again the dye ? Þ I read the figure 11 and understand this at the end of the paper. Use the hybrid obtained in a new dyeing process, i.e. desorbing the dye from the clay in order to reuse it.
That's right, it's just the process you describe.
- Base on the table 1, the adsorptions are above 98% (not 95%). I have some doubts on your ability to measure a concentration of 6.57 10-4gL-1, perhaps 6.610-4 gL-1 Ads% 99.316 : 99.3% ? What are the standard deviation of your measures? (adding “±” would give value to your results)
Desviation has been added
- Table 2 : same remark than table 1 Þ accuracy of the data would be interested…
For the values in table 2 only one measurement has been made and therefore no error can be established. Looking at many other articles this type of measurement is only done once and no error is calculated.
- Line 122 : there is an extra « . »
has been deleted
- Figure 2. Calcinationat which temperature and for how much time? (add: 600°C for 3 hours)
has been added
- 134 : It is assume that the dye becomes fixed between the layers. This is an assumption, it is difficult to check whether the dye in on the surface of the platelets or between the platelets ? It seems that the figure 11 shows that you believe more in a surface bonding than an intercalation. If it is the case, you just write adsorption on the clay in the text…
Indeed, we refer to the process as dye adsorption.
- Figure 3 : calcination temperature and temperature dwell. (add: 600°C for 3 hours)
has been added
- 165 : NO3-
has been modified
- 176 : Nanoclay has a shape memory ? You mean that the bonds between the platelets are strong enough to reaggregate the platelets after exfoliation ?
That is indeed the case.
- 132 :CO32
has been modified
- 183 : formats, acetate(formats ?)
The correct word is formates, it has been changed. You can find support for this information in bibliography reference nº. 45.
- 184 : Azo bond(RN2R’)
has been added
- 245-47 : Glass transition could you give an idea of the temperature of it ? You made a comparison with alloy ? Both the polyester and the dye have a specific glass transition and above the glass transition of poylester (~68°C ?) both phase are mixed at the molecular scale.
has been added: since when they reach the glass transition temperature (~68°C) , dye and textile join in a similar way to that of metals when they are melted. Above the glass transition of poylester both phase are mixed at the molecular scale.
- 272 : surface deposition of the hybrid add: (i.e. adsorption of the hybrid on the fiber) Did I understand well?
That sentence yes, what we want to express is the hypothesis of whether it has only been deposited on the surface of the fibre or whether a dye has been produced. Finally, it is observed that dyeing has indeed taken place.
Reviewer 2 Report
The paper is focused on Dyeing with hydrotalcite hybrid nanoclays and disperse, basic and direct dyes
In Results and Discussion:
In Figures 2 and 3 the y axis representation of the diffractograms of the hydrotalcite without calcining (H) and the hydrotalcite after calcination is not conventional.
The authors should explain why they chose this representation, since the conventional representation in terms of intensity (a.u).
The authors should index the diffractograms with the JCPDS card number of the hydrotalcite.
I suggest that authors show the conventional representation of the digratograms, for better comparison with the JCPDS card number of the hydrotalcite.
In Fourier transform infrared spectroscopy FTIR-ATR analysis the authors represented the FTIR spectra with wavenumber as a function of intensity (a.u.). However, it did not specify whether the measurements were performed as a function of absorbance or transmittance. I suggest that the FTIR spectra be represented as a function of one of these parameters and not the intensity, beyond what it is not correct to put intensity a.u. with numerical scale, as shown in Figure 4.
In Figure 5, the DB199 espectrum shows low resolution, the low intensity bands appear as noisy. I suggest that this figure be redone by modifying the y-axis scale.
Authors should improve the discussion and presentation of Figure 10 (SEM and TEM images).
The authors must investigate the hydrotalcite morphology, used as nanoadsorbent, from BET analysis, determining the surface area, total volume of pores and the mean pore diameter.
Author Response
In Figures 2 and 3 the y axis representation of the diffractograms of the hydrotalcite without calcining (H) and the hydrotalcite after calcination is not conventional.
The authors should explain why they chose this representation, since the conventional representation in terms of intensity (a.u).
the y-axis has been changed to the conventional terms such as intensity (a.u) in figures 2 and 3
The authors should index the diffractograms with the JCPDS card number of the hydrotalcite.
I suggest that authors show the conventional representation of the digratograms, for better comparison with the JCPDS card number of the hydrotalcite.
In section 2.3 was added: Previous studies using XRD analysis of the Mg-Al supports show a typical hydrotalcite structure (2θ = 11.27; 34.46°, JCPDS n° 220700) and the XRD patterns in figure 2 demonstrate the presence of mixed oxides due to the presence of Mg(Al3+)O of the MgO-periclase type in accordance with JCPDS n° 450946 [32]. By observing the traces of the calcined sample, we can see its correspondence with JCPDS n° 211152, which corresponds to the MgAl2O4 spinel structure [33]. Furthermore, XRD patterns of the dried Mg-Al (Figure 2) reveal two distinct crystalline phases: MgO (JCPDS 450946) and hydrotalcite phase (JCPDS 220700).
In Fourier transform infrared spectroscopy FTIR-ATR analysis the authors represented the FTIR spectra with wavenumber as a function of intensity (a.u.). However, it did not specify whether the measurements were performed as a function of absorbance or transmittance. I suggest that the FTIR spectra be represented as a function of one of these parameters and not the intensity, beyond what it is not correct to put intensity a.u. with numerical scale, as shown in Figure 4.
In figures 4 and 5 the absorbance has been specified and the units on the y-axis have been removed for the figure.
In Figure 5, the DB199 espectrum shows low resolution, the low intensity bands appear as noisy. I suggest that this figure be redone by modifying the y-axis scale.
Figure 5 has been modified by smoothing the line to eliminate noise.
Authors should improve the discussion and presentation of Figure 10 (SEM and TEM images).
In section 4.1 Material was added: Within figure 10, image "b" shows how the structure has been destroyed by calcination and in figure "c" it can be seen how the structure has been recovered being very similar to the original "a". In addition, in image "e" it can be seen how the layers are further apart after calcination, which improves the adsorbent capacity as it is more likely that the adsorbate can be introduced between these layers.
The authors must investigate the hydrotalcite morphology, used as nanoadsorbent, from BET analysis, determining the surface area, total volume of pores and the mean pore diameter.
A BET analysis has been performed and the results and comments have been added in section 2.8 “BET surface area and porosity measurements”.
Round 2
Reviewer 2 Report
The suggestions and recommendations were considered by authors and the manuscript was modified accordingly. However, the scale of the y-axis of Figures 2 and 3 must be removed, since the representation is in a.u.
Author Response
Axis and scales in figures 2 and 3 have been deleted.